# The Predatory Properties of Bradymonabacteria, the Representative of Facultative Prey-Dependent Predators

**DOI:** 10.3390/microorganisms12102008

**Published:** 2024-10-03

**Authors:** Shuo Wang, Ya Gong, Guan-Jun Chen, Zong-Jun Du

**Affiliations:** 1School of Life Science, Yantai University, Yantai 264005, China; wangshuomicro@ytu.edu.cn; 2Marine College, Shandong University, Weihai 264209, China; gongya@sdu.edu.cn (Y.G.); guanjun@sdu.edu.cn (G.-J.C.); 3State Key Laboratory of Microbial Technology, Shandong University, Qingdao 266237, China

**Keywords:** bradymonabacteria, facultatively prey-dependent predator, needle-less T3SS*, S-motility, kin discrimination

## Abstract

Bradymonabacteria, as the representative of the facultative prey-dependent predators, were re-classified from the preceding *Deltaproteobacteria* into the phylum *Myxococcota* and proposed as a novel class named *Bradymonadia*. However, it was ambiguous whether their predatory pattern and properties were similar to those of the other myxobacterial predators. Therefore, the physiologic features were compared to determine the similarities and differences during the process of group attack and kin discrimination. Comparative genomic analyses were performed to conclude the core genome encoded commonly by bradymonabacteria, *Myxococcia*, and *Polyangia*. In conclusion, we proposed that bradymonabacteria have a predation pattern similar to the that of the representative of opportunistic predators like *Myxococcus xanthus* but with some subtle differences. Their predation was predicted to be initiated by the needle-less T3SS*, and the S-motility mediated by T4P also participated in the process. Meanwhile, their group attacks relied on cell contact and cell destiny. Inter-species (strains) kin discriminations occurred without the existence of T6SS. However, no extracellular lethal substance was detected in the fermentation liquor culture of bradymonabacteria, and the death of prey cells could only be observed when touched by their cells. Moreover, the prey-selective predation was observed when the predator encountered certain prey from *Bacillus* (G^+^), *Algoriphagus* (G^−^), and *Nocardioides* (G^+^). Bradymonabacteria can be regarded as a potential consumer and decomposer, and preying on many sea-dwelling or human pathogenic bacteria allows this group a broad application prospect in marine culture and clinical disease control. Our study will provide more evidence for its exploitations and applications.

## 1. Introductions

Bradymonabacteria is a new-found predatory bacterial group, first discovered in marine environments by Wang et al., accompanied by the establishment of the order *Bradymonadales* [1]. Bradymonabacteria are characterized by preying on other bacteria, similar to many bacterial predators like *Myxococcus xanthus* [2].

In 2020, Waite et al. reclassified *Deltaproteobacteria* using GTDB and GTDB-Tk software to perform phylogenetic analyses based on the core genome and referring to the biological functions of each bacterial group [3]. After the reclassification, the phylum *Myxococcota* (the previous *Myxococcales*) was established and separated from the previous class *Deltaproteobacteria* [4]. Then, three novel phyla, *Desulfobacterota* (the previous *Desulfobacteria*, including *Thermofobacteria* and candidate phylum *Dadabacteria*), *Bdellovibrionota* (the previous *Bdellovibrionadels*, including *Oligoflexia*), and a phylum containing the SAR324 group, were also independent [3]. However, among the reclassifications above, bradymonabacteria were ignored because the available genomic data of these new-found predatory bacteria were insufficient to support the necessary phylogenomic analyses [3]. Therefore, Chuvochina et al. provided new proof for the re-taxonomy of bradymonabacteria applying GTDB-Tk, and then the new-found predatory group was separated from the class *Deltaproteobacteria* [5]. Once the paper was published, the superlative taxon of bradymonabacteria was the class *Bradymonadia*, parallel with the classes *Myxococcia* and *Polyangia*, belonging to the novel phylum *Myxococcota* [5].

Nevertheless, according to their unique metabolic pathways analyses, Mu et al. suggested that bradymonabacteria are a group of facultative prey-dependent predators [2]. A framework to categorize the current bacterial predators was proposed: (i) obligate predators (completely prey-dependent), such as most of the BALOs; (ii) facultative predators (facultatively prey-dependent), such as the bradymonabacteria; and (iii) opportunistic predators (prey-independent), such as *Myxobacteria* and *Lysobacter* sp. [2]. Therefore, although bradymonabacteria was reclassified into the phylum *Myxococcota*, their predatory pattern and properties should be discussed with the other myxobacterial predators.

To sum up, this study will discuss the differences and similarities of the predation between bradymonabacteria and predatory myxobacteria. To achieve a credible conclusion, further comparative genomics analyses should be performed within the comparisons of physiological features and predatory properties. The predatory pattern of bradymonabacteria will be summarized to achieve the major aim of providing evidence for the future exploration of these predators. Subsequently, further studies of the predation mechanisms and extension of the species resources will be facilitated and promoted by this study.

## 2. Materials and Methods

### 2.1. The Comparisons of the Physiologic Features

Combining the analysis results of the genomic comparison mentioned, a horizontal comparison of the physiologic features of bradymonabacteria and the members of *Myxococcota* was also performed to assist in yielding credible references.

#### 2.1.1. Lethality Tests with Transwells

The specific genes *flgM* and *rubisco* were found according to the genomic information of *Br. sediminis* FA350^T^ and *Algoriphagus marinus* am2^T^. Then, the specific primers and Taqman probes were designed using Integrated DNA Technologies (https://sg.idtdna.com) (Appendix A). The genome DNAs of the predator (OD_600_ = 0.6) and prey (OD_600_ = 1.1) in the exponential phase were extracted with the TaKaRa genome extraction kit, and then genes *flgM* and *rubisco* were amplified (Appendix A). The productions were purified with TaKaRa DNA purification kit and conjoined with pMD18-T in Solution I at 37 °C overnight. The concentration of conjunction was measured and changed into the copy number, and then the production was serially diluted to 10^4^. Then, absolute quantitative PCR was performed with specific primers and Taqman probes using these dilutions and primary production as the templates (Appendix A). The fitting standard curve of C_T_ versus gene copy number was yielded.

The cells of *Br. sediminis* FA350^T^ (chamber b) and *A. marinus* am2^T^ (chamber a) were co-cultured in 0.8 and 4.0 µm Transwells (Appendix A). Meanwhile, mono-cultured predator (chamber b) and prey (chamber a) were selected as the control group. After culturing for 0, 68, and 120 h, the predator and prey cells were collected separately. Then, the genome DNA was extracted from the cells and stored at −80 °C. Finally, absolute quantitative PCR with Taqman probes was performed with these extractions as the templates (Appendix A).

#### 2.1.2. The Social Motility Mediated by Type IV Pili

The motile means of cells at the edge of the colony on NA medium was observed by optical microscope to determine the type of social motion of bradymonabacteria.

#### 2.1.3. Kin Discrimination

The cell suspensions of different species (or strains) of bradymonabacteria with the McFarland standard of about 0.5 were spot-inoculated on NA media for adjacent culture trials between each other. After incubation for 3~5 days, the cells at the edge of the colonies were observed.

### 2.2. Comparative Genomic Analyses

The genomic data of bradymonabacteria and *Myxococcota* (including classes *Myxococcia* and *Polyangia*) were obtained from the NCBI database to compare genome sizes and protein-coding gene numbers. The comparative analyses of the specific genes encoded by the genome list with a grey background were performed based on the annotations from BlastKOALA in KEGG (https://www.genome.jp/kegg/, accessed on 16 May 2024). The gene encoded by over 90.0% of all the genomes with over 10 copies was identified as the member belonging to the core genome set. Meanwhile, the specific gene was defined as that encoded by the members of one group but not the other two groups. The integrality of the pathway was compared with KEGG (https://www.genome.jp/kegg/, accessed on 24 September 2024) on the level of modules. The genes encoded by bradymonabacteria related to the needle-less type III secretion system (T3SS*) were filtered according to the criterion of the NR topHSP similarities over 0.4.

### 2.3. The Analyses of the Predatory Property of Bradymonabacteria

#### 2.3.1. Inter-Predation between Different Species (or Strains) of Bradymonabacteria

The cell suspensions with McFarland standards of about 0.5 of different species (or strains) of bradymonabacteria were inoculated with sterile swabs as the pattern of cross-streaking on NA media. Firstly, one suspension was streaked in the 1st direction, and another was streaked vertically in the 2nd direction after the trace of the 1st direction was completely dried. The inter-predations were observed and recorded after incubating for 3~5 days in optimal conditions until the visible lawn formed.

#### 2.3.2. The Comparative Analyses of the Resistance of Different Prey to *Bradymonas sediminis* FA350^T^

A total of six prey, including *Algoriphagus marinus* am2^T^, *Algoriphagus resistens* NH1^T^, *Nocardioides gilvus* XZ17^T^, *Nocardioides albus* JCM 3185^T^, *Bacillus zeae* SDUM 602039, and *Bacillus subtilis* 168, were chosen for the comparison. The standard curves and growth curves of OD_600_ value versus CFU/mL of the six prey above and the predator were measured. The OD_600_ values of different preys at the same CFU/mL were determined according to the standard curves. The suspensions equal to 3 μL of *B. sediminis* FA350^T^ and six preys were spot-inoculated on the modified MA media with the CFU ratios of 10:1, 1:1, and 1:10 for the adjacent culture. The resistances of different preys to the attack from *B. sediminis* FA350^T^ were observed.

## 3. Results and Discussions

As is well-known, the predatory members of myxobacteria, such as *Myxococcus* spp., are the most representative opportunistic predators, which take the wolf-pack strategy [6]. By the time this paper was published, all twelve culturable members of bradymonabacteria were demonstrated to be bacterial predators [7,8,9]. As two predatory groups having some similarities in predation clustered in the preceding *Deltaproteobacter* before the reclassification, the homology of predation between bradymonabacteria and the predators in the current phylum *Myxococcota* needed to be discussed and proved. Thus, further physiological analyses were performed to yield more credible conclusions, and the comparisons of the genomic data were considered in this study.

### 3.1. Physiological Properties

The physiological properties of the representative members of bradymonabacteria, *Myxococcia*, and *Polyangia* are summed up from the database for comparative analyses in Table 1. Based on previous studies, bradymonabacteria were proposed as the facultatively prey-dependent group because their genomes contain features different from those of obligate or facultative predators [2]. It seems terribly different from most members of *Myxococcia* and *Polyangia* that the habitats of bradymonabacteria were often marine and salty environments but rarely soil, plant peripheral, animal oral or excrement, and so forth (Table 1). This is coincident with the large distinction of the quantity demand of sodium chloride and the other mineral salt during the growth process of bradymonabacteria, *Myxococcia* and *Polyangia*, respectively.

Moreover, the bradymonabacterial cells do not produce fruiting bodies, which differs from most members in *Myxococcia* and *Polyangia* (Table 1). Harvesting fruiting bodies is a common characteristic of myxobacteria, for the resting cells in the fruiting bodies allow the flora to survive adversity easily (Table 1). In contrast, the fruiting body production is not essential for bradymonabacteria, which may be due to its habitat always being a water environment so that the cells do not need to tolerate the dry environment. Then all the culturable members of bradymonabacteria take facultative anaerobic respiration but most members of *Myxococcia* and *Polyangia* are obligate aerobes, except for *Anaeromyxobacteraceae* and *Polyangiaceae*, which are obligate anaerobes and micro-anaerobes, respectively (Table 1).

On the other hand, myxobacteria is famous for its secondary metabolism production. Some members in *Myxococcia* and *Polyangia* excrete substances with antagonism to fungi, bacteria, viruses, and parasites, as well as secondary metabolism production with anticoagulation, anti-inflammatory, antineoplastic, and cytotoxin functions, of which the most representative antibiotic is epothilone (Table 1). However, no functional secondary metabolites have been found in the fermentation products of bradymonabacteria at present. Except for the family *Vulgatibacteraceae* and *Labilitrichaceae*, most members of *Myxococcia* and *Polyangia* prey on other bacteria or fungi, similarly to bradymonabacteria, with a wide predation spectrum (Table 1). Nevertheless, great distinctions seem to exist in the predatory mechanisms between bradymonabacteria and the two myxobacterial groups.

However, the two predatory clusters still have similarities in metabolic features. For instance, the synthetic pathways of multiple amino acids and growth factors were almost deficient in all genomes of bradymonabacteria and *M. xanthus* DK1622^T^. Neither of the synthetic pathways of fatty acids was found (Table 1). The multiple auxotrophs shown by these bacterial predators implied that they could provide superior nutrients for these cells to take up amino acids from the prey by employing predation [2].

### 3.2. Similarities in Predation between Bradymonabacteria and the Predatory Myxobacteria

According to the previous studies, the similarities between bradymonabacteria and predatory myxobacteria are summarized below.

The predation of bradymonabacteria on prey bacteria is contact-dependent, the same as for the predators in *Myxococcota* [6,20,21]. As the results of the lethality trials of these predators show, the supernatant of liquid cultures with no cells could not kill bacterial prey (Figure 1). Once the prey encountered the predators in the Transwell tests, their cell walls broke, and the cell contents discharged. However, the prey survived when prevented from these predators by the 0.4 µm microporous filter (Figure 2 and Appendix A).

Moreover, cells of bradymonabacteria invade prey accompanied by the means of S-motility [9]. S-motility is an important pattern of a group attack, named the wolf-pack, for myxobacteria, that occurs with A-motility (adventurous-). S-motility is usually mediated by the rotation of the type IV pili [6,20,22]. Homological analyses of pilin encoded by members of bradymonabacteria were, therefore, performed using MAFFT with the maximum likelihood method, according to the amino acid sequences of proteins, including the type IVa, IVb, and IVc pili, archaea flagellin, and the type II and IV secretion systems [23,24]. The results indicated that the type IVa pilin of bradymonabacteria was homologous with *M. xanthus* DK1622^T^ [2,25]. Meanwhile, the tad IVb pilin had homology with *Bdellovibrio bacteriovorus* HD100^T^ [2,26].

According to the previous studies, the S-motility demonstrated by the cells of *Persicimonas caeni* YN101^T^ cells was less obvious than that by the other three species [9]. The predator *P. caeni* YN101^T^ encodes the largest genome among the current culturable bradymonabacteria, with more complete metabolic pathways and more multiple copies of repeat genes but fewer predatory properties [9]. To sum up, it can be inferred that the predation of bradymonabacteria may be reflected in their social motilities to some extent. When the cells of these predator groups need more prey cells for nutrients to stay alive, they will try to encounter more by expanding their range of social motilities. Nevertheless, though the cells of *P. caeni* YN101^T^ performed the least predation, they expanded their range of the social motilities to accelerate the extent of predation gradually with the requirement of the prey cells and the prolongation of the culture time [9].

Lastly, the efficiency of the group attack of bradymonabacteria depends widely on the group density of predators, similarly to predatory myxobacteria (Figure 3). However, the nutritive cells of some myxobacteria can form thin biofilms to cooperate with the multiplication and extension of new survival spaces, as well as the invasion of prey [20]. The high density of cells determines the adequate concentrations of extracellular antibiotics and lyases for efficient predation and dissolution [27]. Nevertheless, no bacterial toxicity subtracts were detected in the fermentation liquor of bradymonabacteria, so we predict that the promotion of the efficiency of predation was related to the high frequency of cell contact. Meanwhile, single cells of myxobacteria can still permeate the micro-colony of prey and make them disintegrate until death [28]. A similar mechanism was not demonstrated in the new predatory group.

### 3.3. Specific Features in Predation

The proliferation of the cells of bradymonabacteria depended on the predation to some extent but not completely. This is distinct from the predators of the myxobacteria group, which get rid of the restraint in prey scarcity. For instance, the cells of *Bradymonas sediminis* can grow on a suitable medium (such as an NA medium), but when provided with suitable prey bacteria, they can achieve better growth (Appendix A).

Moreover, there was no lethal compound in the exocellular substance produced by cells of bradymonabacteria according to the results of Transwell and lethality tests of liquid culture (Figure 1 and Figure 2, Appendix A). Meanwhile, no antibiotic compound was detected in the liquid cultures yet. It indicated that the predation occurred until the predator and prey finished touching each other, and we speculated that the adhesion to prey cells before predation is necessary for their group attack strategy. Antibiotics and lyases are excreted by the nutrient cells of myxobacteria when they encounter their prey cells to disintegrate [21]. Then, the wreckages of the prey are digested into small molecular extracellularly and absorbed by the predatory cells [29]. Taking *M. xanthus* as an example, its cells can produce and spread broad-spectrum lethal compounds via cooperation to react to the target in the prey cells, contributing to the rupture and extinction of prey. The cells can also simultaneously respond to the exogenous AHLs produced during the signal transduction among the prey cells nearby [30].

Selective predation usually occurs when bradymonabacteria encounter members of *Bacillus* (G^+^), *Algoriphagus* (G^−^), and *Nocardioides* (G^+^) (Figure 3). Cells of *Bradymonas sediminis* could prey on *Bacillus zeae*, *Algoriphagus marinus*, and *Nocardioides gilvus* but not *Bacillus subtilis*, *Algoriphagus resistens*, and *Nocardioides albus* (Figure 3). The predator behaves as having a subtle prey-preferring mechanism. On the other hand, it may mean that unusual predatory patterns cause the predator to invade different prey in different manners.

### 3.4. Kin Discrimination

The phenomenon named colony-merger incompatibility is an important phenotype of kin discrimination, which allows bacterial individuals to identify self from non-self [31]. When two colonies with the motility characteristics of different strains but the same species encounter each other, kin discrimination occurs between them [32]. If the opposite side recognizes both as self, two colonies will fuse into one; once one is recognized as non-self, a boundary will form between them. Kin discrimination can ensure the coexistence of the different strains in the same habitat to maintain their respective territories and protect their resources from plundering by non-self [31].

Colony-merger incompatibility is often observed among the colonies of myxobacteria, such as *M. xanthus* [29]. The main reason for this kin discrimination is that different strains of *M. xanthus* can combine the nuclease toxin carried by the PAAR protein with the top of the type VI secretion system to transport them to the nearby cells [33]. If the nearby cells are self, the corresponding immune proteins will be produced by then to damage the nuclease toxin; if not, they will be killed by the toxin [33]. Thus, the cells living at each edge of two nonhomologous colonies excrete toxins with different targets so that the visible boundary forms [34].

The adjacent culturing of different bradymonabacteria species (strains) was performed to demonstrate the colony-merger incompatibility that occurred on the border of their colonies (Figure 4). The boundary with the most visibility was observed between the *Persicimonas caeni* YN101^T^ and *Microvenator marinus* V1718^T^ colonies (Figure 4). Nevertheless, although the colonies of *Bradymonas sediminis* FA350^T^ were apparently separated from *Lujinxingia sediminis* SEH01^T^ and *P. caeni* YN101^T^, the edges of the colonies were not explicit enough (Figure 4). An obvious borderline formed from the colonies of *L. sediminis* SEH01^T^ when co-cultured adjacently with *P. caeni* YN101^T^, while the growth of the strain was inhibited during this process (Figure 4). *Br. sediminis* FA350^T^ colonies grew over those of *M. marinus* V1718^T^, with no boundary forming. It could be estimated as a common circumstance that two heterogeneous colonies confront each other according to their size and morphological characteristics (Figure 4). The colony merge shown as a contrast in Figure 4 occurred in the same species of *Br. sediminis* FA350^T^ and *P. caeni* YN101^T^, respectively. However, no potential gene was found to participate in the metabolic pathway of the type VI secretion system based on the genomic annotations of bradymonabacteria [2]. Thus, the mechanism of this colony-merger incompatibility needs further exploration and discussion.

### 3.5. Comparative Genomic Analyses

According to the annotations from the NCBI, the comparative analyses of all available genome sequences (including assembling sequences) and culturable genomes of *Myxococcia* and *Polyangia* were performed (Table 2). The distribution of the genome size and the number of protein-encoding genes are displayed in Figure 5A. The size of the whole bradymonabacterial genome is smaller than that of *Myxococcia* and *Polyangia*. The genome of *Persicimonas caeni* YN101^T^ is of the largest size (8,047,306 bp) among those of bradymonabacteria, and the genome of the *Bradymonadaceae* bacterium (GCA_018609835.1) of 1,909,418 bp assembled from the unculturable metagenomic data is the smallest one [7]. However, the size of most bradymonabacterial genomes is about 5 Mbp. Although the genome encoded by *P. caeni* YN101^T^ is the largest one among available bradymonabacteria, it is still smaller than that of *Corallococcus silvisoli* c25j21 and *Plesiocystis pacifica* SIR-1, of which the sizes are in the quartile among all the members of *Myxococcia* and *Polyangia*, respectively [35,36]. It is interesting that the genome size of *Pajaroellobacter abortibovis* is 1,821,632 bp, which belongs to *Polyangia*, while most of *Myxococcia* and *Polyangia* are of the size from 10 to 11 Mbp [37].

Moreover, the number of proteins encoded by the genomes of bradymonabacteria and the members of *Myxococcia* and *Polyangia* is also demonstrated in Figure 5B,C. The discrepancies in the functions presented by the genomes of bradymonabacteria, *Myxococcia*, and *Polyangia* are demonstrated by the Venn diagram in Figure 5D based on the annotations and analyses via KEGG.

A total of 196 genes are contained by the three groups simultaneously, and there exist 25 genes without known functions, including 15 hypothetical proteins. It was found that highly copied synthetic enzymes and transporters were associated with the metal element molybdenum among the completed annotated genes, such as molybdate transport systems (*modA* and *modB*) [38]. These genes encode the transporters that can import molybdate intracellularly to assist the cells to finish the synthetic pathways needed for molybdenum ions, for instance, the synthesis of tetrahydrofolic acid (THFA) [39]. Moreover, several other multi-copied genes related to the synthesis of THFA were found in the common genes, including the molybdenum purine transferase (*moeA*), molybdenum cofactor guanylate transferase (*mobA*), and MoaE-MoaD fusion protein (*moaX*) [40,41]. In addition, several highly copied genes affiliated with efflux systems were also found among the common genes, especially the efflux systems of multiple drugs (*acrA*, *mexA*, *adeI*, *smeD*, *mtrC*, and *cmeA*), multiple antibiotics (*marC*), and heavy metal ions (*czcA*, *cusA*, *cnrA*, *czcB*, *cusB*, and *cnrB*) [42]. Multi-copies of protein V (*yscV*, *sctV*, *hrcV*, *ssaV*, and *invA*) and protein R (*yscR*, *sctR*, *hrcR*, and *ssaR*) are also encoded by the genomes of *Myxococcia* and *Polyangia*, which are the parts of the type III secretion system components, as well as the transporter of long-chain fatty acids [43]. All the proteins above were considered to affiliate with the special metabolic pathways of bradymonabacteria. For instance, the type III secretion system was predicted to participate in the antibiotic and toxin secretion processes during the predation launched by bradymonabacteria, and the gene *fadL* was proposed as the key to the accumulation of intracellular PHA (Figure 6) [44].

Compared with the members of *Myxococcia* and *Polyangia*, 112 specific genes are encoded by the bradymonabacterial genomes, and 21 genes of uncertain functions exist among them, including 8 hypothetic proteins. It is special that multi-copied sodium/potassium/chloride transporters (*SLC12A2* and *NKCC1*) and manganese/zinc/iron transport system permeases (*troD*, *mntD*, *znuB*, *troC*, *mntC*, *znuB*, *troB*, *mntB*, *znuC*, *troA*, *mntA*, and *znuA*) are contained by the genomes of bradymonabacteria, which is probably to apply their habitat of marine or highly salty environments [45,46]. The genomes of bradymonabacteria encode numerous copies of nitrite reductase (*nirk*) simultaneously, which coincides with the nitrite-reducing functions of most of their members [47].

On the other hand, there are 159 common genes encoded by the genomes of *Myxococcia* and *Polyangia* without bradymonabacteria, including 110 genes with uncertain functions. Notably, the type VI secretion system plays a critical function in the kin recognition of the cells of *Myxococcia* and *Polyangia*, which is lost in the genomes of bradymonabacteria. It is indicated that an undiscovered secretion system encoded by these new predators may exist. Thus, the inter-strain recognition mechanism of bradymonabacteria should be further discussed.

As is well-known, the type IV pili of *M. xanthus* always have the function of the type IV secretion system, which plays an important role in the predation process [48,49]. However, the results of genomic annotations and analyses have indicated that genomes of bradymonabacteria do not encode the entire type IV secretion system (T4SS), which perhaps means its predation occurs without the secretion system [2]. Meanwhile, the type III secretion system (T3SS) and Tad-like system were annotated to be encoded by the genomes of the predators, which may indicate that these two transmembrane structures play an important role during predations (Appendix A) [50]. This is because previous studies proved that the Tad-like system is necessary to induce prey cell death, while the needle-less T3SS* initiates prey lysis [29]. According to the results of transcription analyses, the genes encoding the T3SS inner-membrane protein complex were significantly upregulated during predation [29]. Thus, it implied that the predation of bradymonabacteria may be mediated by needle-less T3SS*. Moreover, according to the genomic information, none of the polypeptides and antibiotics were annotated as secondary mediated products, and no antimicrobial substance was detected and separated from the fermented liquor. Thus, it is reasonable to infer that the needle-less T3SS* of the predator cell participates in transporting lethal components to prey cells to disintegrate them.

### 3.6. Prediction of Predatory Mechanisms

#### 3.6.1. The Inter-Predation among Different Species (Strains) of Bradymonabacteria

The crossing-streak tests demonstrated the inter-predation among different species (strains) of bradymonabacteria (Figure 7).

As Figure 7 shows, *Lujinxingia litoralis* B210^T^ preyed on *M. marinus* V1718^T^ and *P. caeni* YN101^T^. The growth territory of *L. litoralis* B210^T^ invaded, by means of S-motility, the districts of *M. marinus* V1718^T^ and *P. caeni* YN101^T^, as shown in the enlarged picture in Figure 7. According to the analyses of the previous works, the predators in the genus *Lujinxingia* possess more superiority in S-motility than *M. marinus* V1718^T^ and *P. caeni* YN101^T^. This may indicate that the appearance of S-motility of these predators could objectively reflect their need for prey.

#### 3.6.2. The Comparative Analyses of Different Prey Resisting the Predation from *Bradymonas sediminis* FA350^T^

The growth curves of the predator *Br. sediminis* FA350^T^ as well as six prey species were measured, including *Algoriphagus marinus* am2^T^, *Algoriphagus resistens* NH1^T^, *Nocardioides gilvus* XZ17^T^, *Nocardioides albus* JCM 3185^T^, *Bacillus zeae* SDUM 602039, and *Bacillus subtilis* 168. Meanwhile, the standard curves of OD_600_ versus CFU/mL of the liquid cultures of the bacteria above were also fitted to determine the cell concentration in the following tests (Appendix A).

The confront culture trials were performed using the liquid cultures of predator *Br. sediminis* FA350^T^ and six preys with an equal volume. The predatory property was found to be associated with the cell concentration of *Br. sediminis* FA350^T^ (Figure 3). Especially when the cell concentration ratio between the predator and the prey was increased from 1:1 to 10:1, the prey considered to possess anti-predatory ability before, such as *A. resistens* NH1^T^, *N. albus* JCM 3185^T^, and *Ba. subtilis* 168, were killed by *Br. sediminis* FA350^T^ to various degrees. Moreover, as the culture time prolonged, the so-called anti-predatory ability was observed to vanish gradually in some prey. Thus, we predicted that the reversible anti-predation that appeared might depend more on the growth situation of prey.

## 4. Conclusions

To sum up, bradymonabacteria, as the representative of facultative prey-dependent predators, is a predatory group with a predation pattern similar to that of predatory myxobacteria but with some subtle specificity. For instance, their group attacks accompanied by S-motility rely on cell contact and density. The specific predatory features are outstanding. These predators live in salty water environments without functional secondary metabolites produced. In particular, selective predations to certain groups, like *Bacillus* (G^+^), *Algoriphagus* (G^−^), and *Nocardioides* (G^+^), always occur on bradymonabacteria. Meanwhile, interspecies kin discriminations of cells were observed, reflecting their predatory properties indirectly, but their genomes did not encode T6SS. Based on the genomic analyses, we speculated that the needle-less T3SS* may substitute T4SS in initiating predation. Thus, considering the results of their inter-species (strain) predations and resistance from their prey, future study directions could focus on explaining the predation mechanisms by discussing the anti-predation of *Bacillus subtilis*.

Furthermore, bradymonabacteria can be regarded as a potential consumer and decomposer in biogeochemical cycles, for it may regulate the structure of microbial flora. As a functional bacterial group, these predators may play a role in natural selection in the environment, exhibiting extremely critical ecological significance. As a marine-dwelling or salt-needing predatory group, bradymonabacteria is rarely discovered in freshwater environments. However, it has been reported to be found in the intestines of pregnant pigs, which provides a new prospect for exploring the potential habitats and suggests a new direction to enrich the new species resource [51]. Preying on many sea-dwelling or human pathogenic bacteria allows this group a broad application prospect in marine culture and clinical disease control.

## Figures and Tables

**Figure 1 microorganisms-12-02008-f001:**
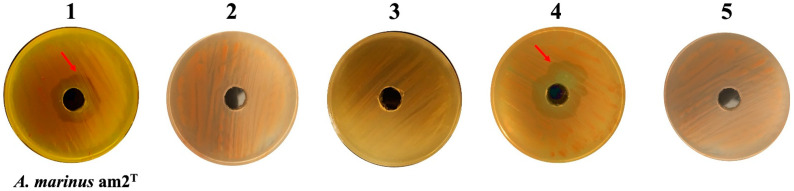
Lethality assays on *Algoriphagus marinus* am2^T^ with fermentation liquid, fermentation supernatant, and cells of *Bradymonas sediminis* FA350^T^. 1, fermentation liquid *Br. sediminis* FA350^T^ cultured with NB medium; 2, supernatant of fermentation liquid culture of *Br. sediminis* FA350^T^ (sterilized with nylon hydrophilic filtration of 0.22 μm); 3, pure NB medium without *Br. sediminis* FA350^T^ cells; 4, cell suspension of *Br. sediminis* FA350^T^ with 4% NaCl solution; 5, 4% NaCl solution.

**Figure 2 microorganisms-12-02008-f002:**
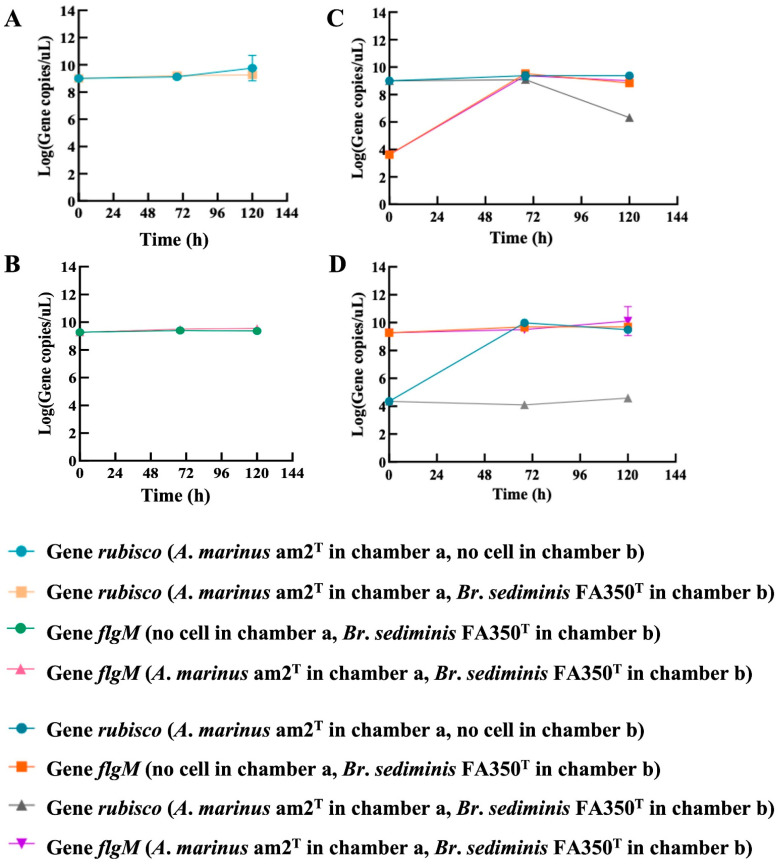
The concentration changes in specific genes encoded by *A. marinus* am2^T^ (gene *rubisco*) and *Br. sediminis* FA350^T^ (gene *flgM*) cultured in chambers a and b of Transwells, separately. (**A**) gene concentration in chamber a of 0.4 μm Transwell; (**B**) gene concentration in chamber b of 0.4 μm Transwell; (**C**) gene concentration in chamber a of 8.0 μm Transwell; (**D**) gene concentration in chamber b of 8.0 μm Transwell.

**Figure 3 microorganisms-12-02008-f003:**
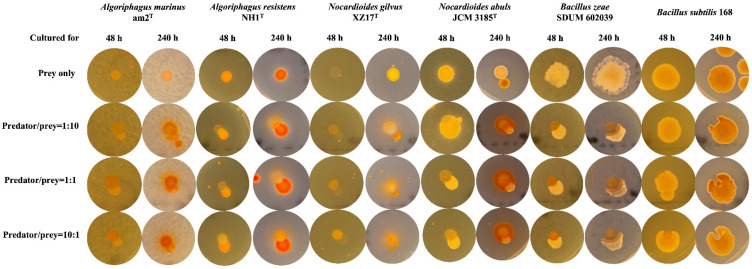
The anti-predation of six prey co-cultured with *Br. sediminis* FA350^T^ in different cell concentration ratios. Each prey was cultured with the predator for 48 h and 120 h to compare the anti-predation. The monocultured preys are shown in the first row. The second to fourth rows show the co-cultured predators and prey inoculated in cell concentration ratios of 1:10, 1:1, and 10:1, separately.

**Figure 4 microorganisms-12-02008-f004:**
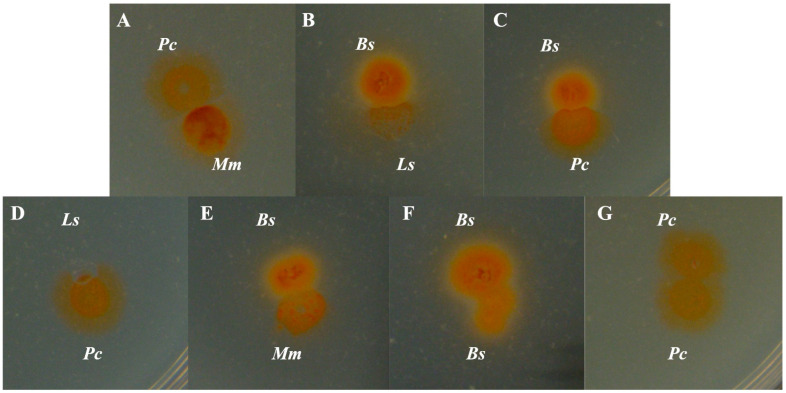
Inter-species (strain) kin discrimination of bradymonabacteria. (**A**) *Persicimonas caeni* YN101^T^ (upper) and *Microvenator marinus* V1718^T^ (lower); (**B**) *Bradymonas sediminis* FA350^T^ (upper) and *Lujinxingia sediminis* SEH01^T^ (lower); (**C**) *Br. sediminis* FA350^T^ (upper) and *P. caeni* YN101^T^ (lower); (**D**) *Lujinxingia sediminis* SEH01^T^ (upper) and *P. caeni* YN101^T^ (lower); (**E**) *Br. sediminis* FA350^T^ (upper) and *M. marinus* V1718^T^ (lower); (**F**) *Br. sediminis* FA350^T^ (upper) and *Br. sediminis* FA350^T^ (lower); (**G**) *P. caeni* YN101^T^ (upper) and *P. caeni* YN101^T^ (lower).

**Figure 5 microorganisms-12-02008-f005:**
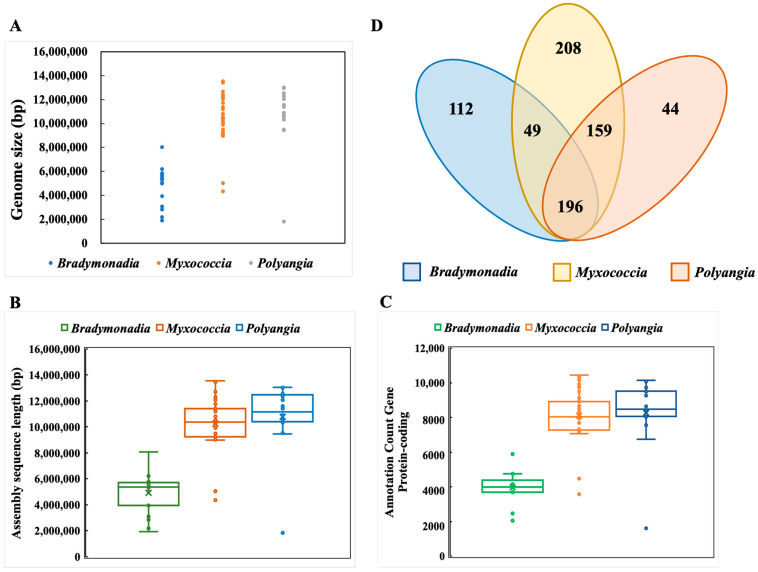
Scatter diagram of the genome size distribution (**A**), box plot of the assembly genome sequence length (**B**), box plot of the protein-encoding gene number (**C**), and Venn diagram of the discrepancies in the functions encoded by genomes (**D**) of *Bradymonadia*, *Myxococcia*, and *Polyangia*.

**Figure 6 microorganisms-12-02008-f006:**
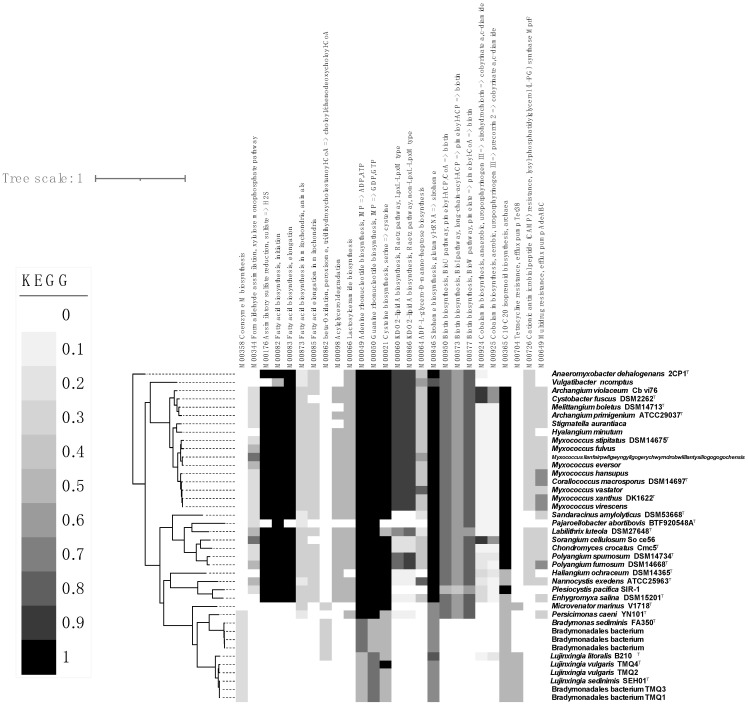
Heatmap of the pathway analyses among *Bradymonadia*, *Myxococcia*, and *Polyangia*. The topologies were clustered using GTDB-tk. The degree of color represents the pathway integrality. The darkest color means the pathway is complete (the parameter was 1.0), and the lightest means no pathway exists (the parameter was 0). The diagram was modified with the iTOL online tools.

**Figure 7 microorganisms-12-02008-f007:**
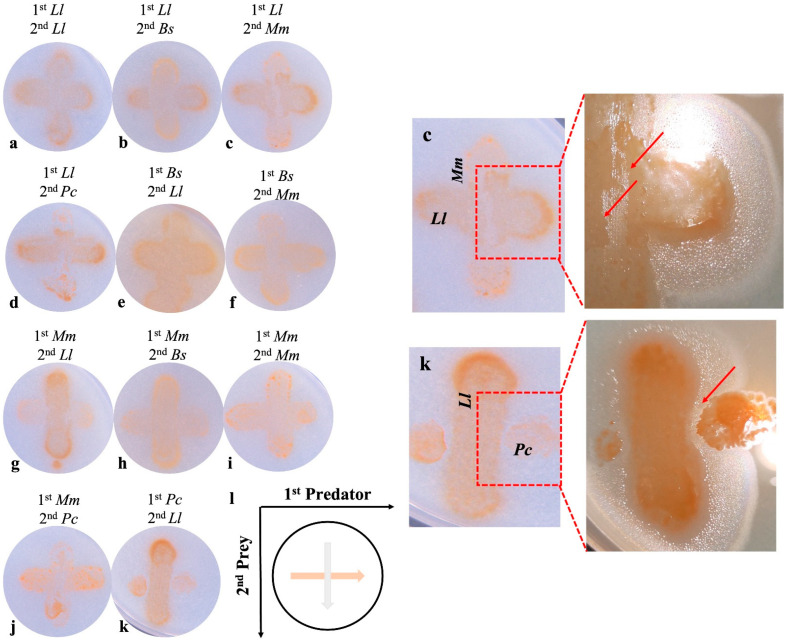
The inter-predation among different species (strains) of bradymonabacteria inoculated by cross-streaking means. (**a**–**k**), cross-streaking culture of different species (strains); (**l**) schematic of crossing-streaking inoculation; (**c**,**k**) are partially enlarged. *Ll*, *Lujinxingia litoralis*; *Bs*, *Bradymonas sediminis*; *Mm*, *Microvenator marinus*; *Pc*, *Percisimonas caeni*.

**Table 1 microorganisms-12-02008-t001:** The physiological characteristics of *Myxococcia*, *Polyangia*, and *Bradymonadia*.

Class	*Myxococcia*				*Polyangia*				*Bradymonadia*		
Order	*Myxococcales* *				*Haliangiales* $	*Nannocystales* §	*Polyangiales* ¶		*Bradymonadales* €		
Family	*Anaeromyxobacteraceae*	*Myxococcaceae*	*Vulgatibacteraceae*	*Labilitrichaceae*	*Haliangiaceae*	*Nannocystaceae*	*Polyangiaceae*	*Sandaracinaceae*	*Bradymonadaceae*	*Lujinxingiaceae*	*Microvenatoraceae*
Habitat	Soil, sediment, water, rock, bark, rotting wood, dung, plant leaf	Soil, freshwater, marine	Forest soil	Forest soil	Soil, dung, bark, marine environment	Soil, saline environment, dung, lake	Terrestrial soil samples containing decaying plant materials	Soil sample containing plant residues	Intertidal zone or marine solar saltern	Intertidal zone	Intertidal zone
Fruiting body	+/−	+ (Except for *Pyxidicoccus*)	−	−	+/−	+	+ (Tree-like structure)	+ (Fruit body-like aggregates)	−	−	−
Growth with oxygen	Strictly Anaerobic growth	Aerobic growth	Aerobic growth	Aerobic growth	Aerobic growth	Aerobic growth	Aerobic growth/Microaerobic growth	Aerobic growth	Facultative anaerobic growth	Facultative anaerobic growth	Facultative anaerobic growth
Stress tolerance	Some degree of salt tolerance	Salt can be tolerated by some marine-derived strains	NA	NA	0.5% NaCl in VY/2 medium	Mostly tolerate higher sodium chloride	Temperature and desiccation resistant	NA	Middle degree of salt tolerance	Middle degree of salt tolerance	Middle degree of salt tolerance
Temperaturefor growth	Mesophilic	Mesophilic	15−30 °C	20−37 °C	Mesophilic	Mesophilic	Mesophilic	Mesophilic	15–50 °C	20–45 °C	20–45 °C
Salt for growth	Low amount	No sodium chloride required	NA	Low amount	Low amount	Halotolerant	Low amount	NA	1–10 NaCl (%, *w*/*v)*	0–8 NaCl (%, *w*/*v*)	1–10 NaCl (%, *w*/*v*)
Nitrate reduction	+/−	NA	+	+	−	NA	NA	NA	+	+/−	+
GC content	68 mol%	70 mol%	66.3−68.3 mol%	66.3 mol%	67−69 mol%	65−72 mol%	69−70 mol%	66.8 mol%	61.1–63.8 mol%	64.1–65.0 mol%	55.2 mol%
Secondary metabolism production	Antifungal, antibiotic, and antitumor activities	Antimicrobials, antiparasitics, antivirals, cytotoxins, and anti-blood coagulants	NA	NA	Haliangiacins	NA	Polyunsaturated fatty acids	Anti-inflammatory, antitumor, and antiviral agents/ Epothilone	NA	NA	NA
Predatory features	Most members prey on *Escherichia coli*	Prey on bacteria and yeasts	Not predate	No vigorous growth observed	Prey on dead or living bacteria or yeasts and grow well	Lyse and prey on bacteria	Most members prey on bacteria	Predators, but yeast *Schizosaccharomyces pombe* and acid-fast *Nocardia flava* not lysed	Lyse and prey on bacteria	Lyse and prey on bacteria	Lyse and prey on bacteria

+: Positive; −: negative; NA, data not available. *****: [10,11,12,13,14]; **$**: [15]; **§**: [16,17]; **¶**: [18,19]; **€**: [1,8,9].

**Table 2 microorganisms-12-02008-t002:** Genomic data of culturable bradymonabacteria and *Myxococcota* (*Myxococcia* and *Polyangia*) used for comparative analyses. The genome list with grey background was used for the comparison of metabolic pathways.

Assembly Name	Assembly Accession	Organism Name	Class	Assembly Stats Total Sequence Length	Annotation Count Gene Protein Coding
ASM1860983v1	GCA_018609835.1	*Bradymonadaceae* bacterium	*Bradymonadia*	1,909,418	
ASM336753v1	GCA_003367535.1	*Bradymonadaceae* bacterium TMQ3	*Bradymonadia*	5,366,856	4003
ASM355957v1	GCA_003559575.1	*Bradymonadales* bacterium	*Bradymonadia*	3,946,580	
ASM356603v1	GCA_003566035.1	*Bradymonadales* bacterium	*Bradymonadia*	2,826,758	
ASM769360v1	GCA_007693605.1	*Bradymonadales* bacterium	*Bradymonadia*	2,185,313	2033
ASM1693153v1	GCA_016931535.1	*Bradymonadales* bacterium	*Bradymonadia*	5,703,155	4085
ASM1838491v1	GCA_018384915.1	*Bradymonadales* bacterium	*Bradymonadia*	3,068,288	2454
ASM799497v1	GCA_007994975.1	*Bradymonadales* bacterium TMQ1	*Bradymonadia*	5,357,521	3952
ASM325831v1	GCF_003258315.1	*Bradymonas sediminis* FA350^T^	*Bradymonadia*	5,045,683	3684
ASM326012v1	GCF_003260125.1	*Lujinxingia litoralis* B210^T^	*Bradymonadia*	5,008,384	3787
ASM400556v1	GCF_004005565.1	*Lujinxingia sediminis* SEH01^T^	*Bradymonadia*	5,329,124	3960
ASM799701v1	GCF_007997015.1	*Lujinxingia vulgaris* TMQ2	*Bradymonadia*	5,553,159	4143
ASM799700v1	GCF_007997005.1	*Lujinxingia vulgaris* TMQ4^T^	*Bradymonadia*	5,588,314	4246
ASM799375v1	GCA_007993755.1	*Microvenator marinus* V1718^T^	*Bradymonadia*	5,847,748	4758
ASM799377v1	GCA_007993775.1	*Persicimonas caeni* YN101^T^	*Bradymonadia*	8,047,206	5892
ASM356283v1	GCA_003562835.1	*Lujinxingiaceae* bacterium	*Bradymonadia*	5,370,514	
ASM356593v1	GCA_003565935.1	*Lujinxingiaceae* bacterium	*Bradymonadia*	5,764,115	
ASM1369763v1	GCA_013697635.1	*Lujinxingiaceae* bacterium	*Bradymonadia*	6,213,341	4748
ASM2214v1	GCF_000022145.1	*Anaeromyxobacter dehalogenans* 2CP-1	*Myxococcia*	5,029,329	4466
ASM1903903v1	GCF_019039035.1	*Citreicoccus inhibens*	*Myxococcia*	9,046,820	7067
ASM361216v1	GCF_003612165.1	*Corallococcus aberystwythensis*	*Myxococcia*	9,981,836	7996
ASM361169v1	GCF_003611695.1	*Corallococcus carmarthensis*	*Myxococcia*	10,793,792	8659
ASM25529v1	GCF_000255295.1	*Corallococcus coralloides* DSM 2259^T^	*Myxococcia*	10,080,619	7981
ASM1311670v1	GCF_013116705.1	*Corallococcus exercitus*	*Myxococcia*	10,256,052	8123
ASM1730297v1	GCF_017302975.1	*Corallococcus exiguus*	*Myxococcia*	10,538,407	8260
ASM366887v1	GCF_003668875.1	*Corallococcus interemptor*	*Myxococcia*	9,471,940	7681
ASM361205v1	GCF_003612055.1	*Corallococcus llansteffanensis*	*Myxococcia*	10,526,383	8311
ASM230589v1	GCF_002305895.1	*Corallococcus macrosporus* DSM 14697^T^	*Myxococcia*	8,973,512	7166
ASM361212v1	GCF_003612125.1	*Corallococcus praedator*	*Myxococcia*	10,510,652	8481
ASM361173v1	GCF_003611735.1	*Corallococcus sicarius*	*Myxococcia*	10,393,104	8043
ASM990914v1	GCF_009909145.1	*Corallococcus silvisoli*	*Myxococcia*	9,232,290	7252
ASM361163v1	GCF_003611635.1	*Corallococcus terminator*	*Myxococcia*	10,351,705	8133
ASM188735v1	GCF_001887355.1	*Cystobacter ferrugineus*	*Myxococcia*	12,051,756	9735
ASM33547v2	GCF_000335475.2	*Cystobacter fuscus* DSM 2262^T^	*Myxococcia*	12,282,374	9829
ASM2010372v1	GCF_020103725.1	*Cystobacter gracilis*	*Myxococcia*	11,731,597	9119
ASM73731v1	GCF_000737315.1	*Hyalangium minutum*	*Myxococcia*	11,186,183	8725
ASM126320v1	GCF_001263205.1	*Labilithrix luteola* DSM27648^T^	*Myxococcia*	12,191,466	10,441
ASM230585v1	GCF_002305855.1	*Melittangium boletus* DSM 14713^T^	*Myxococcia*	9,910,441	7998
ASM1089445v1	GCF_010894455.1	*Myxococcus eversor*	*Myxococcia*	11,394,166	8918
IMG-taxon 2693429886 annotated assembly	GCF_900111765.1	*Myxococcus fulvus*	*Myxococcia*	10,824,667	8427
ASM28092v3	GCF_000280925.3	*Myxococcus hansupus*	*Myxococcia*	9,490,432	7427
ASM663621v1	GCF_006636215.1	*Myxococcus llanfairpwllgwyngyllgogerychwyrndrobwllllantysiliogogogochensis*	*Myxococcia*	12,411,432	9907
ASM33173v1	GCF_000331735.1	*Myxococcus stipitatus* DSM 14675^T^	*Myxococcia*	10,350,586	7940
ASM1089447v1	GCF_010894475.1	*Myxococcus vastator*	*Myxococcia*	8,990,913	7279
IMG-taxon 2619619029 annotated assembly	GCF_900101905.1	*Myxococcus virescens*	*Myxococcia*	9,240,716	7427
ASM1268v1	GCF_000012685.1	*Myxococcus xanthus* DK 1622^T^	*Myxococcia*	9,139,763	7199
ASM1089440v1	GCF_010894405.1	*Pyxidicoccus caerfyrddinensis*	*Myxococcia*	13,434,823	10,154
ASM1293365v1	GCF_012933655.1	*Pyxidicoccus fallax*	*Myxococcia*	13,532,887	10,333
ASM1089443v1	GCF_010894435.1	*Pyxidicoccus trucidator*	*Myxococcia*	12,670,262	9486
IMG-taxon 2693429895 annotated assembly	GCF_900109545.1	*Stigmatella aurantiaca*	*Myxococcia*	9,365,767	7319
IMG-taxon 2693429888 annotated assembly	GCF_900111745.1	*Stigmatella erecta*	*Myxococcia*	9,217,225	7274
ASM2010377v1	GCF_020103775.1	*Stigmatella hybrida*	*Myxococcia*	9,159,828	7247
ASM126317v1	GCF_001263175.1	*Vulgatibacter incomptus*	*Myxococcia*	4,350,553	3570
ASM102728v1	GCF_001027285.1	*Archangium gephyra*	*Polyangia*	12,489,432	9733
OleMiss_Aprim	GCF_016904885.1	*Archangium primigenium* ATCC 29037^T^	*Polyangia*	9,491,554	7558
ASM73329v1	GCF_000733295.1	*Archangium violaceum* Cb vi76	*Polyangia*	12,537,762	10,134
Chondromyces apiculatus DSM 436 genome assembly	GCF_000601485.1	*Chondromyces apiculatus* DSM 436^T^	*Polyangia*	11,577,721	8475
ASM118929v1	GCF_001189295.1	*Chondromyces crocatus* Cmc5^T^	*Polyangia*	11,388,132	8044
ASM299463v1	GCF_002994635.1	*Enhygromyxa salina* DSM 15201^T^	*Polyangia*	10,602,692	8110
ASM2480v1	GCF_000024805.1	*Haliangium ochraceum* DSM 14365^T^	*Polyangia*	9,446,314	6743
ASM234391v1	GCF_002343915.1	*Nannocystis exedens* ATCC 25963^T^	*Polyangia*	12,059,053	9263
ASM2007374v1	GCF_020073745.1	*Nannocystis pusilla*	*Polyangia*	10,899,935	8461
ASM193150v1	GCF_001931505.1	*Pajaroellobacter abortibovis* BTF92054	*Polyangia*	1,821,632	1613
ASM17089v1	GCF_000170895.1	*Plesiocystis pacifica* SIR-1	*Polyangia*	10,587,646	8123
ASM514463v2	GCF_005144635.2	*Polyangium aurulentum*	*Polyangia*	12,320,079	9519
ASM514458v1	GCF_005144585.1	*Polyangium fumosum* DSM 14668^T^	*Polyangia*	12,981,589	10,078
ASM964984v1	GCF_009649845.1	*Polyangium spumosum* DSM 14734^T^	*Polyangia*	10,760,459	8409
ASM73732v2	GCF_000737325.1	*Sandaracinus amylolyticus* DSM 53668^T^	*Polyangia*	10,327,335	8632
ASM6716v1	GCF_000067165.1	*Sorangium cellulosum* So ce56	*Polyangia*	13,033,779	9488

## Data Availability

The original contributions presented in the study are included in the Appendix A, further inquiries can be directed to the corresponding authors.

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
