# Peer review of "The Predatory Properties of Bradymonabacteria, the Representative of Facultative Prey-Dependent Predators"

_microorganisms, 2024, doi:10.3390/microorganisms12102008_

Round 1

Reviewer 1 Report

Comments and Suggestions for Authors

In this manuscript, physiological and genomic comparisons of the bacteria Bradymonabacteria, Myxococcia, and Polyangia were performed. The authors present solid evidence of their analyses at different levels and have a strong relationship with the description of the genes encoded by the Bradymonabacteria and their predicted functions. Due to the above, I consider that the manuscript could use a slight restructuring to describe the comparative genomics section, as well as writing details.

1. The authors performed the comparative genomics analysis of 70 genomes (¿?) described in the supplementary material 2 mentioned in the results. From this, I have the doubt of what the X in the column of that table means. Also, I consider that this information (the genomes used) should be placed in materials and methods since your results are based on the comparison of these genomes. For this, you should place a table that includes the genomes with the genome data (in the comparative genomics section, for example) instead of presenting it as supplementary material.

2. Regarding this section, you mention that performed an annotation using the KEGG database “The annotations of the genomic data of culturable bradymonabacteria and Myxococcota were performed through the KEGG database for comparative analyses”. You need to be more descriptive, indicating the software that allowed them to perform this annotation, as well as the parameters used.

3. For the identification of the core genes and the specific genes, what was the bioinformatics strategy for the clustering? Under what percentage of identity and other parameters did you perform the gene clusters? Please include it in materials and methods since you are presenting a very scarce form of information, an important approach that you relied on in the discussions and results.

4, Up to what class or level of KEGG did you perform the gene comparison? Is it possible to present it in a diagram or image that shows the genomes and KEGG levels analyzed in order to visualize it quickly for the reader?

Others:

Line 108: Bradymonas sediminis please write it in italics and standardize the names of bacteria in this way throughout your manuscript. Same observation for lines 398 - 400.

Figure 5. You use Bradymonabacteria with italics (B and C) and without italics (A and D) please standardize the style that you will use in your images

Line 310: “bradymonabaterial genomes”, please change to bradymonabacterial genomes

Author Response

Comments 1: The authors performed the comparative genomics analysis of 70 genomes (¿?) described in the supplementary material 2 mentioned in the results. From this, I have the doubt of what the X in the column of that table means. Also, I consider that this information (the genomes used) should be placed in materials and methods since your results are based on the comparison of these genomes. For this, you should place a table that includes the genomes with the genome data (in the comparative genomics section, for example) instead of presenting it as supplementary material.

Response 1: Thank you for pointing this out. We agree with this comment. We have moved Supplementary Material 2 to materials and methods named Table 2. Moreover, we are very sorry that we neglected to delete the X column in previous Supplementary Material 2, which contained the reference parameters used when building plots to distinguish three groups. We have deleted this column in the new Table 2. (Table 2)

Comments 2: Regarding this section, you mention that performed an annotation using the KEGG database “The annotations of the genomic data of culturable bradymonabacteria and Myxococcota were performed through the KEGG database for comparative analyses”. You need to be more descriptive, indicating the software that allowed them to perform this annotation, as well as the parameters used.

Response 2: Agree. We have, accordingly, modified the relevant descriptions to emphasize this point. (Lines 102~112)

Comments 3: For the identification of the core genes and the specific genes, what was the bioinformatics strategy for the clustering? Under what percentage of identity and other parameters did you perform the gene clusters? Please include it in materials and methods since you are presenting a very scarce form of information, an important approach that you relied on in the discussions and results.

Response 3: Agree. We have supplemented this part in the section of materials and methods. The gene encoded by more than 90.0 % of all the genomes with over 10 copies was identified as the member belonging to the core genome set. Meanwhile, the specific gene was defined as that encoded by the members of one group but not the other two groups. (Lines 102~112)

Comments 4: Up to what class or level of KEGG did you perform the gene comparison? Is it possible to present it in a diagram or image that shows the genomes and KEGG levels analyzed in order to visualize it quickly for the reader?

Response 4: Thank you for pointing this out. We performed the gene comparison on the lever of modules, and we have present as a diagram in Figure 6. (Lines 102~112, Figure 6)

Comments 5:

Line 108: Bradymonas sediminis please write it in italics and standardize the names of bacteria in this way throughout your manuscript. Same observation for lines 398 - 400.

Figure 5. You use Bradymonabacteria with italics (B and C) and without italics (A and D) please standardize the style that you will use in your images

Line 310: “bradymonabaterial genomes”, please change to bradymonabacterial genomes

Response 5:

Thanks. We have corrected all the clerical error about the bacterial name. (Lines 420~422)

Thanks. We have corrected the format error. (Figure 5)

Thanks. We have corrected the clerical error. (Lines 322~323)

Reviewer 2 Report

Comments and Suggestions for Authors

-          This paper correspond for scope of journal.

-          The title corresponds to the content of the paper. 

 -          The contribution of this work is study of , bradymonabacteria specific characteristics,  as the representative of facultative prey-dependent predators, which live in salty water environments without functional secondary metabolites produced. 

 -          Also, the contribution is that established that predatory property was found to be associated with the cell concentration of Br. sediminis FA350T. Especially when the cell concentration ratio between the predator and the prey was increased from 1:1 to 10:1, the prey considered to possess anti-predatory ability before, such as A. resistens NH1T, N. albus JCM 3185T, and Ba. subtilis 168, were killed by Br. sediminis FA350T to various degrees.

-          Conducted comparative genomic analyses  of all available genome sequences (including assembling sequences) and culturable genomes of Myxococcia and Polyangia were performed and established that the size of the whole bradymonabacterial genome is smaller than that of Myxococcia and Polyangia.

 -          This study addressed to determine the similarities and differences between novel class named Bradymondia and other myxobacterial predators on the base comparison analysis of their physiologic features during the process of group attack and kin discrimination.

 -          Also, study addressed to establish prediction of predatory mechanisms among different species (strains) of bradymonabacteria and establishing predatory property in different cell concentration ratio between the predator and the prey was increased from 1:1 to 10:1, and estimation of level of anti-predatory ability A. resistens NH1T, N. albus JCM 3185T, and Ba. subtilis 168, were killed by Br. sediminis FA350T

 -          The aim is not clearly pointed out in paper. Should be write as a particular last paragraph  of chapter Introduction.

 -          Key words are appropriately chosen!

 -          Scientific methodology is described in detail. Applied methods are relevant and adequate for this investigation. Presented description of methods are useful for researcher in this area.

-          Results are clearly presented and discussed.

 -          Tables, figures, pictures are clear.

 -          The conclusion chapter was written on the base of obtained results.

 -          Manuscript is acceptable after minor corrections.

Sugestion:

In line 311 need make correction bradymonabaterial (wrong) and write bradymonabacterial (right) 

 In line 311 need make correction braydmonabateria (wrong) and write bradymonabacteria (right) 

.I suggest that the authors check the text of the paper and correct other possible typos

Author Response

Comments 1: The aim is not clearly pointed out in paper. Should be write as a particular last paragraph of chapter Introduction.

Response 1: Thank you for pointing this out. We agree with this comment. Therefore, we have modified the main aim of this study as a particular last paragraph of chapter Introduction. (Lines 60~67)

Comments 2: In line 311 need make correction bradymonabaterial (wrong) and write bradymonabacterial (right) 

 In line 311 need make correction braydmonabateria (wrong) and write bradymonabacteria (right) 

Response 2: Agree. We have corrected the clerical error. (Lines 322~323)

Moreover, we have checked the manuscript to avoid the clerical and format error.
